# Hazard Gradient Penalty for Survival Analysis

## Abstract

Survival analysis appears in various fields such as medicine, economics, engineering, and business. Recent studies showed that the Ordinary Differential Equation (ODE) modeling framework integrates many existing survival models while the framework is flexible and widely applicable. However, naively applying the ODE framework to survival analysis problems may model fiercely changing density function with respect to covariates which may worsen the model's performance. Though we can apply L1 or L2 regularizers to the ODE model, their effect on the ODE modeling framework is barely known. In this paper, we propose *hazard gradient penalty* (HGP) to enhance the performance of a survival analysis model. Our method imposes constraints on local data points by regularizing the gradient of hazard function with respect to the data point. Our method applies to any survival analysis model including the ODE modeling framework and is easy to implement. We theoretically show that our method is related to minimizing the KL divergence between the density function at a data point and that of the neighborhood points. Experimental results on three public benchmarks show that our approach outperforms other regularization methods.

## 1 Introduction

Survival analysis (a.k.a time-to-event modeling) is a branch of statistics that predicts the duration of time until an event occurs (Kleinbaum & Klein, 2012). Survival analysis appears in various fields such as medicine (Schwab et al., 2021), economics (Meyer, 1988), engineering (O'Connor & Kleyner, 2011), and business (Jing & Smola, 2017; Li et al., 2021). Due to the presence of right-censored data, which is data whose event has not occurred yet, survival analysis models require special considerations. Cox proportional hazard model (CoxPH) (Cox, 1972; Katzman et al., 2018) and accelerated time failure model (AFT) (Wei, 1992) are widely used to handle right-censored data. Yet the assumptions made by these models are frequently violated in the real world (Lee et al., 2018; Tang et al., 2022a). Recent studies showed that the Ordinary Differential Equation (ODE) modeling framework integrates many existing survival analysis models including CoxPH and AFT (Groha et al., 2020; Tang et al., 2022a;b). They also showed that the ODE modeling framework is flexible and widely applicable.

However, naively applying the ODE framework to survival analysis problems may result in wildly oscillating density function that may worsen the model's performance. Regularization techniques that can regularize this undesirable behavior are understudied. Though applying L1 or L2 regularizers to the ODE model is one option, their effects on the ODE modeling framework are barely known. The cluster assumption from semi-supervised learning states that the decision boundaries should not cross high-density regions (Chapelle et al., 2006). Likewise, survival analysis models need hazard functions that slowly change in high-density regions.

Suppose we attempt to predict the time to death of three individuals A, B, and C. Assume the traits of A and B are similar and the traits of B and C are dissimilar. It is natural to expect that the probability distribution of time-to-death of A should be close to that of B while far from that of C. The expectation aligns with the cluster assumption. Explicitly modeling the assumption enhances the performance as long as it holds.

In this paper, we propose hazard gradient penalty to make a slowly changing (with respect to co-variates) survival analysis model in high-density regions. In a nutshell, the hazard gradient penalty

regularizes the gradient of the hazard function with respect to the data point from the real data distribution. Our method has several advantages. 1) The method is computationally efficient. 2) The method is theoretically sound. 3) The method is applicable to any survival analysis model including the ODE modeling framework as long as it models hazard function. 4) It is easy to implement. We theoretically show that our method is related to minimizing the KL divergence between the density function at a data point and that of the neighborhood points of the data point.

Experimental results on three public benchmarks show that our approach outperforms other regularization methods.

## 2 PRELIMINARIES

Survival analysis data comprises of an observed covariate $\boldsymbol{x}$, a failure event time $t$, and an event indicator $e$. If an event is observed, $t$ corresponds to the duration time from the beginning of the follow-up of an individual until the event occurs. In this case, the event indicator $e = 1$. If an event is unobserved, $t$ corresponds to the duration time from the beginning of follow-up of an individual until the last follow-up. In this case, we cannot know the exact time of the event occur and event indicator $e = 0$. An individual is said to be *right-censored* if $e = 0$. The presence of *right-censored* data differentiates survival analysis from regression problems. In this paper, we only focus on the single-risk problem where event $e$ is a binary-valued variable.

Given a set of triplet $\mathcal{D} = \{(\boldsymbol{x}_i, t_i, e_i)\}_{i=1}^N$, the goal of survival analysis is to predict the likelihood of an event occur $p(t \mid \boldsymbol{x})$ or the survival probability $S(t \mid \boldsymbol{x})$. The likelihood and the survival probability have the following relationship:

$$S(t \mid \boldsymbol{x}) = 1 - \int_0^t p(\tau \mid \boldsymbol{x})d\tau \tag{1}$$

Modeling $p(t \mid \boldsymbol{x})$ or $S(t \mid \boldsymbol{x})$ should satisfy the following constraints:

$$p(t \mid \boldsymbol{x}) > 0, \quad \int_0^\infty p(\tau \mid \boldsymbol{x})d\tau = 1$$

$$S(0 \mid \boldsymbol{x}) = 1, \quad \lim_{t \to \infty} S(t \mid \boldsymbol{x}) = 0, \quad S(t_1 \mid \boldsymbol{x}) \geq S(t_2 \mid \boldsymbol{x}) \text{ if } t_1 \leq t_2$$

Previous works instead modeled the hazard function (a.k.a conditional failure rate) $h(t \mid \boldsymbol{x})$ (Cox, 1972; Katzman et al., 2018; Wei, 1992; Zhong et al., 2021).

$$h(t \mid \boldsymbol{x}) := \lim_{\Delta t \to 0} \frac{P(t \leq T < t + \Delta t \mid T \geq t, \boldsymbol{x})}{\Delta t} = \frac{p(t \mid \boldsymbol{x})}{S(t \mid \boldsymbol{x})} \tag{2}$$

As the hazard function is a probability per unit time, it is unbounded upwards. Hence, the only constraint of the hazard function is that the function is non-negative: $h(t \mid \boldsymbol{x}) \geq 0$

### 2.1 THE ODE MODELING FRAMEWORK

We can obtain an ODE which explains the relationship between the hazard function and the survival function by putting derivative of equation 1 into equation 2 (Kleinbaum & Klein, 2012).

$$h(t \mid \boldsymbol{x}) = \frac{p(t \mid \boldsymbol{x})}{S(t \mid \boldsymbol{x})} = \frac{1}{S(t \mid \boldsymbol{x})}\left(-\frac{dS(t \mid \boldsymbol{x})}{dt}\right) = -\frac{d\log S(t \mid \boldsymbol{x})}{dt} \tag{3}$$

Starting from initial value $\log S(0 \mid \boldsymbol{x}) = 0$, we can define $\log S(t \mid \boldsymbol{x})$ as the solution of the ODE initial value problem where the ODE is defined as equation 3 [1].

$$\log S(t \mid \boldsymbol{x}) = \log S(0 \mid \boldsymbol{x}) + \int_0^t -h(\tau \mid \boldsymbol{x})d\tau = \int_0^t -h(\tau \mid \boldsymbol{x})d\tau$$

---

[1]Tang et al. (2022b)'s formulation is slightly different in that their hazard function also depends on the cumulative hazard. To our understanding, depending on cumulative hazard is redundant so we conduct experiments without it.

We can train the ODE model by minimizing the negative log-likelihood.

$$\mathcal{L}_{\boldsymbol{x}} = -e \log p_\theta(t \mid \boldsymbol{x}) - (1-e) \log S_\theta(t \mid \boldsymbol{x}) \tag{4}$$
$$= -e \left(\log h_\theta(t \mid \boldsymbol{x}) + \log S_\theta(t \mid \boldsymbol{x})\right) - (1-e) \log S_\theta(t \mid \boldsymbol{x})$$

Following Groha et al. (2020), we update the model parameters using Neural ODEs (Chen et al., 2018). The hazard function $h_\theta(t \mid \boldsymbol{x})$ is modeled using a neural network followed by the `softplus` activation function to ensure that the output is always non-negative.

## 2.2 NEURAL ODEs

Neural ODEs model the continuous dynamics of variables (Chen et al., 2018). Starting from $\boldsymbol{z}(0)$, we can define the output $\boldsymbol{z}(T)$ to be the solution of the following ordinary differential equation (ODE) initial value problem.

$$\frac{d\boldsymbol{z}(t)}{dt} = f(\boldsymbol{z}(t), t, \theta), \quad \boldsymbol{z}(T) = \boldsymbol{z}(0) + \int_0^T f(\boldsymbol{z}(t), t, \theta) dt$$

Naively applying an ODE solver to an ODE initial value problem leads to practical difficulties. An ODE solver builds a big computation graph which incurs high memory cost and additional numerical errors may occur in backpropagation steps. Chen et al. (2018) showed that we can obtain the gradients of a scalar-valued loss w.r.t all inputs of any ODE solver with constant memory cost. We can calculate the gradients without backpropagating through the operations of the solver but with another call to an ODE solver.

## 3 METHODS

In this section, we introduce the hazard gradient penalty and show that it is related to minimizing the KL divergence between the density function at a data point and that of its neighbours. See Figure 1 for the graphical overview of our method.

The cluster assumption from semi-supervised learning states that the decision boundaries should not cross high-density regions (Chapelle et al., 2006). In a similar vein, hazard functions of survival analysis models should change slowly in high-density regions.

Consider a case where two data points $\boldsymbol{x}_1, \boldsymbol{x}_2 \in \mathbb{R}^d$ failed at $t_1, t_2 (t_1 < t_2)$ each. Under the cluster assumption, a point $\boldsymbol{x}_2' \in B(\boldsymbol{x}_2, \epsilon)^2$ should fail at $t_2' \approx t_2$. If $\hat{p}(t \mid \boldsymbol{x}_2')$ is skewed for some reason and puts high density at $t_1' < t_1$, it worsens the model's performance. To evade such situation, $\hat{p}(t \mid \boldsymbol{x}_2')$ should not deviate too much from $\hat{p}(t \mid \boldsymbol{x}_2)$. To achieve this, we propose the following regularizer.[3]

$$\mathcal{R}_{\boldsymbol{x}} = \mathbb{E}_{t \sim S_\theta(t \mid \boldsymbol{x})} \left[ \|\nabla_{\boldsymbol{x}} h_\theta(t \mid \boldsymbol{x})\|_2 \right] \tag{5}$$

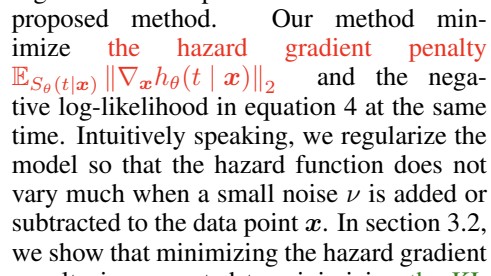

Figure 1: Graphical overview of our proposed method. Our method minimize the hazard gradient penalty $\mathbb{E}_{S_\theta(t \mid \boldsymbol{x})} \|\nabla_{\boldsymbol{x}} h_\theta(t \mid \boldsymbol{x})\|_2$ and the negative log-likelihood in equation 4 at the same time. Intuitively speaking, we regularize the model so that the hazard function does not vary much when a small noise $\nu$ is added or subtracted to the data point $\boldsymbol{x}$. In section 3.2, we show that minimizing the hazard gradient penalty is connected to minimizing the KL divergence between the density at $\boldsymbol{x}$ and the density at $\boldsymbol{x}' \in B(\boldsymbol{x}, \epsilon)$.

---

[2] $B(\boldsymbol{x}, \epsilon)$ is a $\epsilon$-ball centered at $\boldsymbol{x}$

[3] In practice, we implement $h_\theta(t \mid \boldsymbol{x})$ using a neural network whose input is a combination (concatenation, addition or both) of $t$ and $\boldsymbol{x}$. Hence we can write $h_\theta(t \mid \boldsymbol{x})$ and $h_\theta(t, \boldsymbol{x})$ interchangeably. The gradient $\nabla_{\boldsymbol{x}} h_\theta(t, \boldsymbol{x})$ is naturally defined and so is $\nabla_{\boldsymbol{x}} h_\theta(t \mid \boldsymbol{x})$.

### 3.1 EFFICIENT SAMPLING FROM THE SURVIVAL DENSITY

The sampling operation $t \sim S(t \mid \boldsymbol{x})^4$ in equation 5 may induce computational overhead. To boost the sampling operation, we use $\log S_\theta(t \mid \boldsymbol{x})$ which was computed during the negative log-likelihood calculation in equation 4. Let $[t_1, \ldots, t_K]$ be the union of the time points in minibatch. The time points are sorted in increasing order. The adaptive time stepping in ODE solvers are sensitive to the time interval $t_K - t_1$ rather than the number of time points (Rubanova et al., 2019). We can access $\log S_\theta(t_k \mid \boldsymbol{x})$ with negligible overhead as long as $t_1 < t_k < t_K$.

We sample $t_k$ from a categorical distribution whose $k$-th weight is defined as $S_\theta(t_k \mid \boldsymbol{x}) = \exp(-\int_0^{t_k} h(\tau \mid \boldsymbol{x})d\tau)$. We finalize the sampling process by sampling $t$ from the uniform distribution $\mathcal{U}([t_k, t_{k+1}])$. In this way, we don't have to calculate $S(t \mid \boldsymbol{x})$ again for sampling $t$. See Algorithm 1 in Appendix for the pseudo code of ODE based survival analysis with the hazard gradient penalty.

### 3.2 CONNECTION TO KL DIVERGENCE

We now show that the hazard gradient penalty in equation 5 is equivalent to minimizing the approximation of the upper bound of the KL divergence between the density function at a data point and that of the neighborhood points of the data point. Henceforth, we denote $\mathcal{X}$ by the subset of $d$-dimensional real space $\mathbb{R}^d$.

**Theorem 1** *Suppose the hazard function is strictly positive function for all data point $\boldsymbol{x}, \boldsymbol{x}' \in \mathcal{X}$. The KL divergence*

$$\mathbb{E}_{p(t|\boldsymbol{x})}\left[\log p(t \mid \boldsymbol{x}) - \log p(t \mid \boldsymbol{x}')\right]$$

*is upper bounded by*

$$\mathbb{E}_{p(t|\boldsymbol{x})}\left\|\log h(t \mid \boldsymbol{x}) - \log h(t \mid \boldsymbol{x}')\right\|_2 + \mathbb{E}_{S(t|\boldsymbol{x})}\left\|h(t \mid \boldsymbol{x}) - h(t \mid \boldsymbol{x}')\right\|_2 \quad (6)$$

To prove Theorem 1, we need the following lemma.

**Lemma 1** *The expectation of survival densities difference under the density is the negative of the expectation of hazard functions difference under the survival density. In other words,*

$$\mathbb{E}_{p(t|\boldsymbol{x})}\left[\log S(t \mid \boldsymbol{x}) - \log S(t \mid \boldsymbol{x}')\right] = -\mathbb{E}_{S(t|\boldsymbol{x})}\left[h(t \mid \boldsymbol{x}) - h(t \mid \boldsymbol{x}')\right]$$

*for all $\boldsymbol{x}, \boldsymbol{x}' \in \mathcal{X}$*

Proof) We use the fact that $\mathbb{E}_{S(t|\boldsymbol{x})}\left(\log S(t \mid \boldsymbol{x}) - \log S(t \mid \boldsymbol{x}')\right)$ is constant with respect to $t$.

$$\frac{d}{dt}\mathbb{E}_{S(t|\boldsymbol{x})}\left(\log S(t \mid \boldsymbol{x}) - \log S(t \mid \boldsymbol{x}')\right)$$
$$= \frac{d}{dt}\int S(t \mid \boldsymbol{x})\left(\log S(t \mid \boldsymbol{x}) - \log S(t \mid \boldsymbol{x}')\right)dt$$
$$= -\int p(t \mid \boldsymbol{x})\left(\log S(t \mid \boldsymbol{x}) - \log S(t \mid \boldsymbol{x}')\right)dt + \int S(t \mid \boldsymbol{x})\left(-h(t \mid \boldsymbol{x}) + h(t \mid \boldsymbol{x}')\right)dt = 0$$

Hence,

$$\mathbb{E}_{p(t|\boldsymbol{x})}\left[\log S(t \mid \boldsymbol{x}) - \log S(t \mid \boldsymbol{x}')\right] = -\mathbb{E}_{S(t|\boldsymbol{x})}\left[h(t \mid \boldsymbol{x}) - h(t \mid \boldsymbol{x}')\right] \blacksquare$$

---

[4] $S(t \mid \boldsymbol{x})$ is not a valid probability distribution as we cannot guarantee $\int S(t \mid \boldsymbol{x})dt = 1$. Rigorously, we sample $t \sim s(t \mid \boldsymbol{x})$ where $s(t \mid \boldsymbol{x}) = S(t \mid \boldsymbol{x})/\int S(t \mid \boldsymbol{x})dt$. We use $t \sim S(t \mid \boldsymbol{x})$ for notational simplicity. See Appendix C for the existence of $s_\theta(t \mid \boldsymbol{x})$.

We now go back to Theorem 1 and prove the theorem.

$$\mathbb{E}_{p(t|\boldsymbol{x})}\left[\log p(t \mid \boldsymbol{x}) - \log p(t \mid \boldsymbol{x}')\right]$$

$$= \left\|\mathbb{E}_{p(t|\boldsymbol{x})}\left[\log p(t \mid \boldsymbol{x}) - \log p(t \mid \boldsymbol{x}')\right]\right\|_2 \;\; (\because D_{KL} \geq 0)$$

$$= \left\|\mathbb{E}_{p(t|\boldsymbol{x})}\left[\log h(t \mid \boldsymbol{x}) - \log h(t \mid \boldsymbol{x}')\right] - \mathbb{E}_{p(t|\boldsymbol{x})}\left[\log S(t \mid \boldsymbol{x}) - \log S(t \mid \boldsymbol{x}')\right]\right\|_2 \;\; (\because \text{equation 2})$$

$$= \left\|\mathbb{E}_{p(t|\boldsymbol{x})}\left[\log h(t \mid \boldsymbol{x}) - \log h(t \mid \boldsymbol{x}')\right] + \mathbb{E}_{S(t|\boldsymbol{x})}\left[h(t \mid \boldsymbol{x}) - h(t \mid \boldsymbol{x}')\right]\right\|_2 \;\; (\because \text{Lemma 1})$$

$$\leq \left\|\mathbb{E}_{p(t|\boldsymbol{x})}\left[\log h(t \mid \boldsymbol{x}) - \log h(t \mid \boldsymbol{x}')\right]\right\|_2 + \left\|\mathbb{E}_{S(t|\boldsymbol{x})}\left[h(t \mid \boldsymbol{x}) - h(t \mid \boldsymbol{x}')\right]\right\|_2 \;\; (\because \text{triangle inequality})$$

$$\leq \mathbb{E}_{p(t|\boldsymbol{x})}\left\|\log h(t \mid \boldsymbol{x}) - \log h(t \mid \boldsymbol{x}')\right\|_2 + \mathbb{E}_{S(t|\boldsymbol{x})}\left\|h(t \mid \boldsymbol{x}) - h(t \mid \boldsymbol{x}')\right\|_2 \; \blacksquare$$

**Theorem 2** *An approximation of the upper bound of the KL divergence given in equation 6 is upper bounded by $2\epsilon\mathbb{E}_{S(t|\boldsymbol{x})}\left\|\nabla_{\boldsymbol{x}}h(t \mid \boldsymbol{x})\right\|_2$ if $\boldsymbol{x}'$ is in the epsilon ball centered at $\boldsymbol{x}$, i.e. $\boldsymbol{x}' \in \mathcal{B}(\boldsymbol{x}, \epsilon)$.*

To prove the theorem, we first find the approximation.

**Lemma 2** $2\epsilon\mathbb{E}_{S(t|\boldsymbol{x})}\left\|\nabla_{\boldsymbol{x}}h(t \mid \boldsymbol{x})^T(\boldsymbol{x}' - \boldsymbol{x})\right\|_2$ *is an approximation of the upper bound of the KL divergence which is given in equation 6.*

$$\mathbb{E}_{p(t|\boldsymbol{x})}\left\|\log h(t \mid \boldsymbol{x}) - \log h(t \mid \boldsymbol{x}')\right\|_2 + \mathbb{E}_{S(t|\boldsymbol{x})}\left\|h(t \mid \boldsymbol{x}) - h(t \mid \boldsymbol{x}')\right\|_2$$

$$\approx \mathbb{E}_{p(t|\boldsymbol{x})}\left\|\nabla_{\boldsymbol{x}}\log h(t \mid \boldsymbol{x})^T(\boldsymbol{x}' - \boldsymbol{x})\right\|_2 + \mathbb{E}_{S(t|\boldsymbol{x})}\left\|\nabla_{\boldsymbol{x}}h(t \mid \boldsymbol{x})^T(\boldsymbol{x}' - \boldsymbol{x})\right\|_2$$

$$(\because \log h(t \mid \boldsymbol{x}') \approx \log h(t \mid \boldsymbol{x}) + \nabla_{\boldsymbol{x}}\log h(t \mid \boldsymbol{x})^T(\boldsymbol{x}' - \boldsymbol{x})$$

$$\text{and } h(t \mid \boldsymbol{x}') \approx h(t \mid \boldsymbol{x}) + \nabla_{\boldsymbol{x}}h(t \mid \boldsymbol{x})^T(\boldsymbol{x}' - \boldsymbol{x}))$$

$$= \mathbb{E}_{p(t|\boldsymbol{x})}\left\|\frac{\nabla_{\boldsymbol{x}}h(t \mid \boldsymbol{x})^T}{h(t \mid \boldsymbol{x})}(\boldsymbol{x}' - \boldsymbol{x})\right\|_2 + \mathbb{E}_{S(t|\boldsymbol{x})}\left\|\nabla_{\boldsymbol{x}}h(t \mid \boldsymbol{x})^T(\boldsymbol{x}' - \boldsymbol{x})\right\|_2$$

$$= \int \frac{p(t \mid \boldsymbol{x})}{h(t \mid \boldsymbol{x})}\left\|\nabla_{\boldsymbol{x}}h(t \mid \boldsymbol{x})^T(\boldsymbol{x}' - \boldsymbol{x})\right\|_2 dt + \mathbb{E}_{S(t|\boldsymbol{x})}\left\|\nabla_{\boldsymbol{x}}h(t \mid \boldsymbol{x})^T(\boldsymbol{x}' - \boldsymbol{x})\right\|_2 \;\; (\because h(t \mid \boldsymbol{x}) > 0)$$

$$= 2\mathbb{E}_{S(t|\boldsymbol{x})}\left\|\nabla_{\boldsymbol{x}}h(t \mid \boldsymbol{x})^T(\boldsymbol{x}' - \boldsymbol{x})\right\|_2 \; \blacksquare$$

Obviously, $2\mathbb{E}_{S(t|\boldsymbol{x})}\left\|\nabla_{\boldsymbol{x}}h(t \mid \boldsymbol{x})^T(\boldsymbol{x}' - \boldsymbol{x})\right\|_2 \leq 2\mathbb{E}_{S(t|\boldsymbol{x})}\max_{\boldsymbol{x}' \in \mathcal{X}}\left\|\nabla_{\boldsymbol{x}}h(t \mid \boldsymbol{x})^T(\boldsymbol{x}' - \boldsymbol{x})\right\|_2$. As we assumed $\boldsymbol{x}' \in \mathcal{B}(\boldsymbol{x}, \epsilon)$ in Theorem 2, $\max_{\boldsymbol{x}' \in \mathcal{X}}\left\|\nabla_{\boldsymbol{x}}h(t \mid \boldsymbol{x})^T(\boldsymbol{x}' - \boldsymbol{x})\right\|_2$ is achieved when $\boldsymbol{x}' - \boldsymbol{x} = \epsilon\nabla_{\boldsymbol{x}}h(t \mid \boldsymbol{x})/\left\|\nabla_{\boldsymbol{x}}h(t \mid \boldsymbol{x})\right\|_2$. Hence,

$$2\mathbb{E}_{S(t|\boldsymbol{x})}\left\|\nabla_{\boldsymbol{x}}h(t \mid \boldsymbol{x})^T(\boldsymbol{x}' - \boldsymbol{x})\right\|_2 \leq 2\epsilon\mathbb{E}_{S(t|\boldsymbol{x})}\left\|\nabla_{\boldsymbol{x}}h(t \mid \boldsymbol{x})\right\|_2$$

and this concludes the proof. $\blacksquare$

Theorem 2 shows that regularizating the hazard gradient penalty in equation 5 is equivalent to minimizing the approximation of the upper bound of the KL divergence $\mathbb{E}_{p(t|\boldsymbol{x})}[\log p(t \mid \boldsymbol{x}) - \log p(t \mid \boldsymbol{x}')]$. To incorporate the regularizer into the negative log-likelihood loss, we minimize the Lagrange multiplier defined as the sum of the negative log-likelihood and the hazard gradient penalty regularizer.

$$\mathcal{L} = \mathbb{E}_{(\boldsymbol{x},t,e)\sim\mathcal{D}}\left[\mathcal{L}_{\boldsymbol{x}} + \lambda\mathcal{R}_{\boldsymbol{x}}\right] \tag{7}$$

Here, $\lambda$ is a coefficient that balances the negative log-likelihood and the regularizer. See Appendix B for the code snippet of our JAX implementation (Bradbury et al., 2018).

Minimizing the hazard gradient penalty in equation 5 has two advantages over minimizing the KL divergence directly: a) computational efficiency and b) reduced burden of hyperparameter tuning. To compute the KL divergence, we first sample $\boldsymbol{x}' \in B(\boldsymbol{x}, \epsilon)$. We then need to compute four values: $h(t|\boldsymbol{x}), S(t|\boldsymbol{x}), h(t|\boldsymbol{x}')$ and $S(t|\boldsymbol{x}')$. In this case, we have to compute hazard values of every $t \sim S(t|\boldsymbol{x})$. Further, we need one more hazard function integration $S(t \mid \boldsymbol{x}') = \exp(-\int h(t \mid \boldsymbol{x}'))$. On the other hand, regularizing the hazard gradient penalty only need to calculate the gradient of the hazard function.

When it comes to regularizing the KL divergence, we have to set the appropriate value of the regularizing coefficient $\lambda'$ and the size of the ball $\epsilon$. On the other hand, if we regularize the hazard gradient penalty, we don't need to tune $\epsilon$ as $\lambda$ in equation 5 incorporates $\epsilon$.

## 4 EXPERIMENTS

In this section, we experimentally show that *the hazard gradient penalty* outperforms other regularizers. Further, we check the hyperparameter sensitivity of *hazard gradient penalty*. Throughout the experiments, we use three public datasets: Study to Understand Prognoses Preferences Outcomes and Risks of Treatment (SUPPORT) [5], the Molecular Taxonomy of Breast Cancer International Consortium (METABRIC) [6], and the Rotterdam tumor bank and German Breast Cancer Study Group (RotGBSG) [7]. Table 4 summarizes the statistics of the datasets. See Appendix A for evaluation metrics and experimental details.

### 4.1 METHODS COMPARED

We compare *ODE + HGP* with four methods: *vanilla ODE, ODE + L1, ODE + L2 ODE + LCI*. *Vanilla ODE* minimizes the expectation of the negative log-likelihood in equation 4. *ODE + L1* minimizes the Lagrange multiplier defined as the sum of the expectation of the negative log-likelihood and the L1 penalty term: $\mathbb{E}_{(\boldsymbol{x},t,e)\sim\mathcal{D}}\mathcal{L}_{\boldsymbol{x}} + \alpha \sum_{p=1}^{P} |w_p|$

Here, $w_p$s are model parameters and $\alpha$ is a coefficient that balances the negative log-likelihood and the L1 penalty term. This is an extension of Lasso-Cox (Tibshirani, 1997) to the ODE modeling framework. *ODE + L2* minimizes the Lagrange multiplier defined as the sum of the expectation of the negative log-likelihood and the L2 penalty term: $\mathbb{E}_{(\boldsymbol{x},t,e)\sim\mathcal{D}}\mathcal{L}_{\boldsymbol{x}} + \alpha \sum_{p=1}^{P} w_p^2$

Here, $w_p$s are model parameters and $\alpha$ is a coefficient that balances the negative log-likelihood and the L2 penalty term. This is an extension of Ridge-Cox (Verweij & Van Houwelingen, 1994) to the ODE modeling framework. *ODE + LCI* minimizes the Lagrange multiplier defined as the sum of the expectation of the negative log-likelihood and the negative of the lower bound of a simplified version of time-dependent C-index. The regularizer is defined as

$$-\sum_t \frac{\sum_{i=1}^{N}\sum_{j=1}^{N} e_i I(T_i < T_j, T_i < t)(1 + (\log \sigma(S_\theta(t \mid \boldsymbol{x}_i) < S_\theta(t \mid \boldsymbol{x}_j))/\log 2)}{\sum_{i=1}^{N}\sum_{j=1}^{N} e_i I(T_i < T_j, T_i < t)}$$

This is equivalent to time dependent concordance index in Section A.1.1 if we don't take the Kaplan-Meier estimator into account. The regularizer is a reminiscent of the lower bound of C-index (Steck et al., 2007). Although the lower bound of C-index was originally proposed as a substitute of the negative log-likelihood, Chapfuwa et al. (2018) used the lower bound (Steck et al., 2007) as a regularizer of the AFT model (Wei, 1992).

### 4.2 RESULTS

Table 1 shows the $mC^{td}, mAUC$, and $iNBLL$ scores [8]. The hazard gradient penalty outperforms other methods across almost all metrics and datasets. The interesting point is that both L1 and L2 penalties do not affect the ODE model's performance in most cases. We speculate that regularizing the weight norm is effective in CoxPH as the model is simple and has a strong assumption that the hazard rate is constant. On the contrary, regularizing the norm of the weight may not be able to affect the ODE model's performance as ODE models are much more complex than CoxPH. Also, the experimental results highlight the possibility that the performance of the survival analysis models is more related to the local information such as the gradient at each data point rather than the global information such as the weight norm of the model. Figure 2 shows that the *ODE + HGP* effectively regularized the variation of the density with respect to the input while other methods could not.

Table 1 also shows that regularizing the lower bound of the C-index is not effective in many cases. We conjecture that the method is ineffective as the ODE modeling framework is flexible and optimizing the negative log-likelihood can discriminate each data point's rank. Furthermore, regularizing the lower bound of the C-index does not harness the information of neighbors of data points. We also compare HGP against a Neural ODEs specific regularizer. See Appendix D for details.

---

[5] https://github.com/autonlab/auton-survival/blob/master/dsm/datasets/support2.csv

[6] https://github.com/jaredleekatzman/DeepSurv/tree/master/experiments/data/metabric

[7] https://github.com/jaredleekatzman/DeepSurv/tree/master/experiments/data/gbsg

[8] See Appendix A.1 for the details of $mC^{td}, mAUC$, and $iNBLL$.

Table 1: Experimental Results on three datasets. Averages and standard deviations of 7 different random seeds for each setting are shown. See Appendix A.2 for the details of experimental setups. The dagger mark indicates that the result is statistically significant ($p < 0.05$) compared to the result of vanilla ODE.

(a) $mC^{td}(\uparrow)$

| Method | SUPPORT | METABRIC | RotGBSG |
|---|---|---|---|
| CoxPH | $0.672 \pm 0.008$ | $0.649 \pm 0.012$ | $0.710 \pm 0.007$ |
| DeepHit | $0.762 \pm 0.004$ | $0.688 \pm 0.010$ | $0.711 \pm 0.009$ |
| ODE | $0.771 \pm 0.003$ | $0.695 \pm 0.008$ | $0.718 \pm 0.005$ |
| ODE + L1 | $0.771 \pm 0.002$ | $0.697 \pm 0.008$ | $0.715 \pm 0.006$ |
| ODE + L2 | $0.770 \pm 0.006$ | $0.695 \pm 0.006$ | $0.716 \pm 0.005$ |
| ODE + LCI | $0.771 \pm 0.003$ | $0.698 \pm 0.002$ | $0.716 \pm 0.005$ |
| ODE + HGP | $\mathbf{0.775 \pm 0.004}^{\dagger}$ | $\mathbf{0.702 \pm 0.009}$ | $\mathbf{0.723 \pm 0.006}$ |

(b) $mAUC(\uparrow)$

| Method | SUPPORT | METABRIC | RotGBSG |
|---|---|---|---|
| CoxPH | $0.706 \pm 0.010$ | $0.685 \pm 0.008$ | $0.739 \pm 0.009$ |
| DeepHit | $0.800 \pm 0.004$ | $0.720 \pm 0.012$ | $0.741 \pm 0.010$ |
| ODE | $0.810 \pm 0.002$ | $0.729 \pm 0.005$ | $0.746 \pm 0.006$ |
| ODE + L1 | $0.810 \pm 0.002$ | $0.729 \pm 0.005$ | $0.742 \pm 0.006$ |
| ODE + L2 | $0.809 \pm 0.005$ | $0.728 \pm 0.005$ | $0.742 \pm 0.006$ |
| ODE + LCI | $0.810 \pm 0.002$ | $0.731 \pm 0.003$ | $0.743 \pm 0.006$ |
| ODE + HGP | $\mathbf{0.814 \pm 0.002}^{\dagger}$ | $\mathbf{0.732 \pm 0.005}^{\dagger}$ | $\mathbf{0.753 \pm 0.005}^{\dagger}$ |

(c) $iNBLL(\downarrow)$

| Method | SUPPORT | METABRIC | RotGBSG |
|---|---|---|---|
| CoxPH | $0.564 \pm 0.025$ | $0.474 \pm 0.005$ | $0.530 \pm 0.005$ |
| DeepHit | $0.519 \pm 0.004$ | $0.515 \pm 0.008$ | $0.531 \pm 0.006$ |
| ODE | $0.516 \pm 0.015$ | $0.472 \pm 0.005$ | $0.530 \pm 0.012$ |
| ODE + L1 | $0.515 \pm 0.015$ | $0.470 \pm 0.005$ | $0.533 \pm 0.013$ |
| ODE + L2 | $0.517 \pm 0.017$ | $0.470 \pm 0.004$ | $0.537 \pm 0.010$ |
| ODE + LCI | $0.516 \pm 0.015$ | $\mathbf{0.469 \pm 0.002}$ | $\mathbf{0.528 \pm 0.010}$ |
| ODE + HGP | $\mathbf{0.506 \pm 0.011}^{\dagger}$ | $0.479 \pm 0.003$ | $0.530 \pm 0.003$ |

Table 2: Experimental Results on SUPPORT ablated in terms of sample size $M$. We set $\lambda = 10$.

| Method | $mC^{td}(\uparrow)$ | $mAUC(\uparrow)$ | $iNBLL(\downarrow)$ |
|---|---|---|---|
| No reg. | $0.771 \pm 0.003$ | $0.810 \pm 0.002$ | $0.516 \pm 0.015$ |
| $M = 1$ | $\mathbf{0.775 \pm 0.004}$ | $\mathbf{0.814 \pm 0.002}$ | $\mathbf{0.505 \pm 0.010}$ |
| $M = 5$ | $\mathbf{0.775 \pm 0.004}$ | $\mathbf{0.814 \pm 0.002}$ | $\mathbf{0.506 \pm 0.011}$ |
| $M = 10$ | $\mathbf{0.775 \pm 0.004}$ | $\mathbf{0.815 \pm 0.002}$ | $\mathbf{0.505 \pm 0.009}$ |

Table 2 shows the results by varying the number of samples $M$ in the sampling process $t \sim S(t \mid \boldsymbol{x})$ in equation 5. As long as the regularizer is applied, the number of samples $M$ does not affect the performance. Even when $M = 1$, the regularizer works well. Figure 3 shows the results on SUPPORT and RotGBSG datasets by varying the coefficient $\lambda$ in equation 7. Since the performance variation by $\lambda$ is stable, the hyperparameter $\lambda$ can be tuned without much difficulty in practical setups.

Table 3 shows the time taken for training/evaluation step for each methods. The time it takes to train *ODE + L1* and to train *ODE + L2* is slightly faster than vanilla ODE training time. It is straightforward to see that the L1 and L2 penalty makes smooth dynamics as the number of function

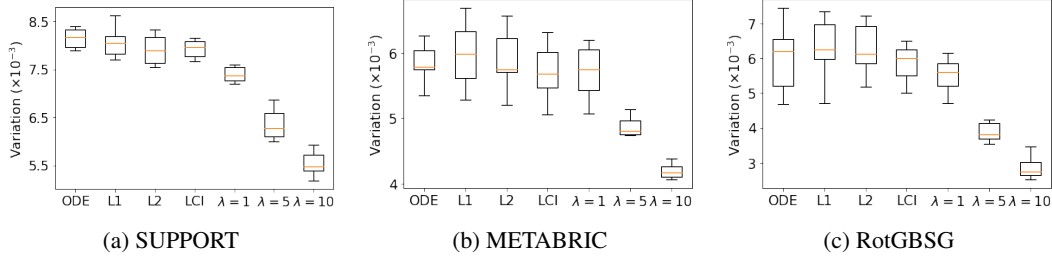

|  |  |  |
|:--:|:--:|:--:|
| (a) SUPPORT | (b) METABRIC | (c) RotGBSG |

Figure 2: A boxplot of the log-likelihood variation with respect to the input perturbation $\mathbb{E}_{(\boldsymbol{x},t)\sim\mathcal{D}_{e=1}}\|\log p_\theta(t\mid\boldsymbol{x}) - \log p_\theta(t\mid\boldsymbol{x}')\|_2$ on three datasets. We denote $\mathcal{D}_{e=1}$ by the set of uncensored data. We choose $\boldsymbol{x}' = \boldsymbol{x} + \epsilon\boldsymbol{g}/\|\boldsymbol{g}\|_2$ where $\boldsymbol{g} = \nabla_{\boldsymbol{x}}\log p_\theta(t\mid\boldsymbol{x})$ so that $\|\log p_\theta(t\mid\boldsymbol{x}) - \log p_\theta(t\mid\boldsymbol{x}')\|_2$ is maximized under $\boldsymbol{x}' \in \mathcal{B}(\boldsymbol{x},\epsilon)$ constraint. We set $\epsilon = 1e-2$ across all experiments. The *hazard gradient penalty* effectively regularizes the variation. As the density of probability distribution $p(\cdot\mid\boldsymbol{x})$ should be concentrated at $t$ for uncensored data $(\boldsymbol{x},t)$, the figures show that the *hazard gradient penalty* effectively regularizes the KL divergence between the density function at a point $\boldsymbol{x}$ and that of neighborhood points $\boldsymbol{x}' \in \mathcal{B}(\boldsymbol{x},\epsilon)$.

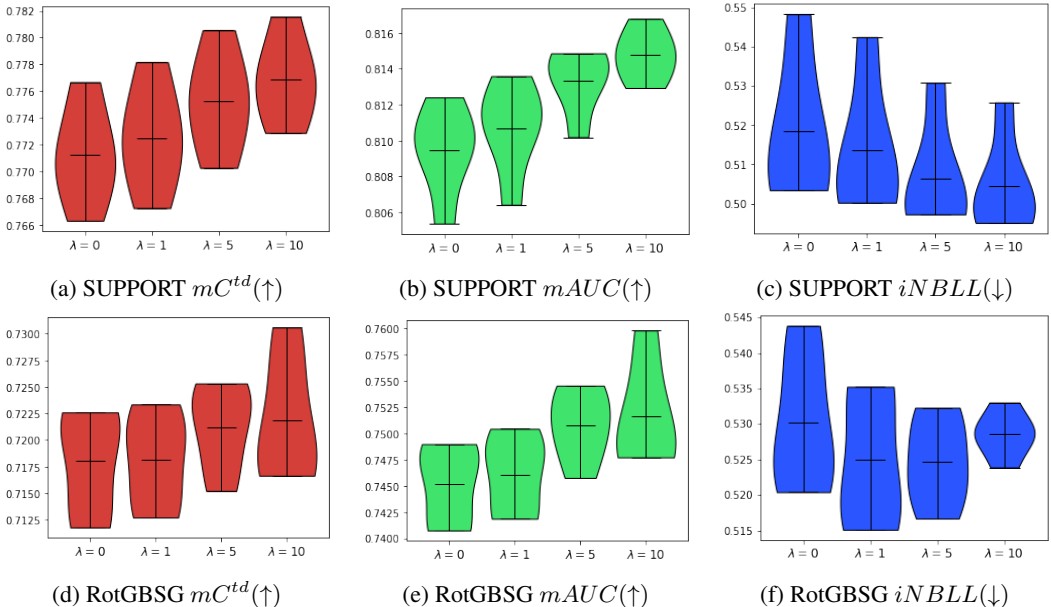

|  |  |  |
|:--:|:--:|:--:|
| (a) SUPPORT $mC^{td}(\uparrow)$ | (b) SUPPORT $mAUC(\uparrow)$ | (c) SUPPORT $iNBLL(\downarrow)$ |
| (d) RotGBSG $mC^{td}(\uparrow)$ | (e) RotGBSG $mAUC(\uparrow)$ | (f) RotGBSG $iNBLL(\downarrow)$ |

Figure 3: Violin plots of experimental results on SUPPORT and RotGBSG by varying $\lambda$. Red, green, and blue denote $mCtd$, $mAUC$, and $iNBLL$. The thickness of a plot denotes the probability density of the results. The *hazard gradient penalty* may conflict with the negative log-likelihood if we set high $\lambda$. The $\lambda$ that achieves the best scores across all metrics on the RotGBSG dataset could have been acquired between $\lambda = 5$ and $\lambda = 10$. However, we report the result at $\lambda = 5$ on the RotGBSG dataset in Table 1c for consistency.

evaluations (NFE) in the training time of *ODE + L1* and *ODE + L2* are lower than that of vanilla ODE. The decrease in NFE compensates for the overheads of calculating L1 and L2 penalties, which makes the training time of *ODE + L1* and *ODE + L2* faster than that of vanilla ODE. The same applies to *ODE + HGP*. Despite the additional sampling process and gradient calculation, the training time of *ODE + HGP* is on par with vanilla ODE. The decreased NFE thanks to smooth dynamics made by the *hazard gradient penalty* compensates for the additional computations. The smooth dynamics made by L1, L2, and the *hazard gradient penalty* can also be observed in the evaluation phase. The NFE of *ODE + L1*, *ODE + L2*, and *ODE + HGP* is much smaller than that of vanilla ODE. Not all regularizers make smooth dynamics. The NFE of *ODE + LCI* is comparable to the NFE of vanilla ODE. Clearly, there's no point in regularizing LCI in terms of inference speed.

Table 3: The taken time (in seconds) and the number of function evaluations (NFE) for each step in the training/evaluation time on RotGBSG. The numbers in parenthesis indicate relative performance against vanilla ODE. HGP incurs negligible overhead (1% slowdown) on training time while gives rise to 10% speedup on evaluation time. See Table 6 for the taken time and the NFE for each step of regularizers on SUPPORT.

|  | ODE | ODE + L1 | ODE + L2 | ODE + LCI | ODE + HGP |
|---|---|---|---|---|---|
| train time | 0.1163 (1) | 0.1043 (0.90) | 0.1125 (0.97) | 0.1173 (1.01) | 0.1179 (1.01) |
| eval time | 0.0020 (1) | 0.0016 (0.80) | 0.0014 (0.71) | 0.0019 (0.97) | 0.0016 (0.80) |
| train NFE | 13.207 (1) | 11.519 (0.87) | 10.637 (0.80) | 13.207 (1.00) | 12.129 (0.91) |
| eval NFE | 11.893 (1) | 9.946 (0.83) | 8.411 (0.70) | 11.643 (0.97) | 9.929 (0.85) |

## 5 RELATED WORKS

A line of research integrated deep neural networks to CoxPH (Faraggi & Simon, 1995; Katzman et al., 2018) and Extended Hazards (Zhong et al., 2021) for more model flexibility. Another line of research proposed distribution-free survival analysis models via the time domain discretization (Lee et al., 2018), adversarial learning approach (Chapfuwa et al., 2018), or derivative-based models (Danks & Yau, 2022). Previous works (Goldstein et al., 2020; Han et al., 2021) proposed new objectives to optimize Brier score (Graf et al., 1999), Binomial log-likelihood, or distributional calibration directly. Yet to the best of our knowledge, none of the previous works focused on the effect of gradient penalty on survival analysis models.

Previous works proposed L1 and L2 regularization in the survival analysis literature (Tibshirani, 1997; Verweij & Van Houwelingen, 1994). Those methods regularize the survival analysis models so that the L1 or L2 norm of the model parameters does not increase so much. Our method is different from those methods in that we penalize the norm of the gradient on each local data point.

Our method is closely related to semi-supervised learning (Chapelle et al., 2006). Among many semi-supervised learning methods, our method is germane to virtual adversarial training (Miyato et al., 2018) in that it regularizes function variation between a local data point and its neighbours. However, virtual adversarial training is different from ours in that the method was demonstrated in the classification setting and the output is a discrete distribution.

In Generative Adversarial Nets (GANs) literature (Goodfellow et al., 2014), the gradient penalty had been studied actively. Gulrajani et al. (2017) proposed the gradient penalty to satisfy the 1-Lipschitz function constraint in Kantrovich-Rubinstein duality. Mescheder et al. (2018) proposed the gradient penalty to penalize the discriminator for deviating from the Nash equilibrium. Ours is different from these works in that we propose gradient penalty so that the density at $x$ does not deviate much from that of $x$'s neighborhood points.

## 6 CONCLUSION

In this paper, we introduced a novel regularizer for survival analysis. Unlike previous methods, we focus on individual local data point rather than global information. We theoretically showed that regularizing the norm of the gradient of hazard function with respect to the data point is related to minimizing the KL divergence between the data point and that of its neighbours. Empirically, we showed that the proposed regularizer outperforms other regularizers and it is not sensitive to hyperparameters. Furthermore, the proposed regularizer is computationally efficient and incurs an ignorable overhead. Nonetheless, as minimizing the proposed regularizer may conflict with optimizing the negative log-likelihood, practitioners should tune the balancing coefficient $\lambda$ for each dataset. The paper highlights the new possibility that the recent advancements in semi-supervised learning could enhance the performance of survival analysis models.

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

Table 4: Summary statistics of the datasets used in our experiments. $N$ denotes the number of data points and $d$ denotes the dimension of each data points.

| Dataset | $N$ | $d$ | Censoring (%) | Durations | | Event Quantiles | | |
| --- | --- | --- | --- | --- | --- | --- | --- | --- |
| | | | | # unique | domain | $t = 25\%$ | $t = 50\%$ | $t = 75\%$ |
| SUPPORT | 9105 | 43 | 31.89% | 1724 | $\mathbb{N}^+$ | 14 | 58 | 252 |
| METABRIC | 1904 | 9 | 42.06% | 1686 | $\mathbb{R}^+$ | 42.68 | 85.86 | 145.33 |
| RotGBSG | 2232 | 7 | 43.23% | 1230 | $\mathbb{R}^+$ | 13.61 | 24.01 | 40.32 |

---

**Algorithm 1** Hazard Gradient Penalty

---

**Require:** $h_\theta$, learning rate $\gamma$
  **repeat**
    sample $(\boldsymbol{x}, t, e) \sim \mathcal{D}$
    Retrieve unique times $t_1, \ldots, t_K$ from minibatch.
    integrate $-h_\theta(t \mid \boldsymbol{x})$ from 0 to $t_K$ and store $\log S_\theta(t_1 \mid \boldsymbol{x}), \ldots, \log S_\theta(t_K \mid \boldsymbol{x})$
    $\log S_\theta(t \mid \boldsymbol{x}) \leftarrow$ choose from $\log S_\theta(t_1 \mid \boldsymbol{x}), \ldots, \log S_\theta(t_K \mid \boldsymbol{x})$ that corresponds to $(\boldsymbol{x}, t, e)$
    $\log p_\theta(t \mid \boldsymbol{x}) \leftarrow \log h_\theta(t \mid \boldsymbol{x}) + \log S_\theta(t \mid \boldsymbol{x})$
    $\mathcal{L}_{\boldsymbol{x}} \leftarrow -e \log p_\theta(t \mid \boldsymbol{x}) - (1 - e) \log S_\theta(t \mid \boldsymbol{x})$                  $\triangleright$ Negative log-likelihood
    sample $i_1, \ldots, i_M \sim \text{Categorical}(S_\theta(t_1 \mid \boldsymbol{x}), \ldots, S_\theta(t_K \mid \boldsymbol{x}))$       $\triangleright$ $t \sim S_\theta(t \mid \boldsymbol{x})$
    $t'_m \leftarrow t_{i_m - 1} + \text{Uniform}(t_{i_m - 1}, t_{i_m})$                               $\triangleright$ $t_0 = 0$
    $\mathcal{R}_{\boldsymbol{x}} \leftarrow \frac{1}{M} \sum_{m=1}^{M} \|\nabla_{\boldsymbol{x}} h_\theta(t'_m \mid \boldsymbol{x})\|_2$         $\triangleright$ Hazard gradient penalty
    $\theta \leftarrow \theta - \gamma \nabla_\theta(\mathcal{L}_{\boldsymbol{x}} + \lambda \mathcal{R}_{\boldsymbol{x}})$
  **until** Convergence

---

# A    EVALUATION METRICS AND EXPERIMENTAL DETAILS

## A.1    EVALUATION METRICS

Throughout this subsection, we denote $\hat{S}(t \mid \boldsymbol{x})$ as the estimate of $S(t \mid \boldsymbol{x})$, $I(\cdot)$ as the indicator function, $(\boldsymbol{x}_i, T_i, e_i)$ as the $i$th covariate, time, event indicator of the dataset, $\hat{G}(t)$ as the Kaplan-Meier estimator for censoring distribution (Kaplan & Meier, 1958), and $\omega_i$ as $1/\hat{G}(T_i)$.

### A.1.1    TIME DEPENDENT CONCORDANCE INDEX ($C^{td}$)

The concordance index, or C-index is defined as the proportion of correctly ordered pairs among all comparable pairs. We use time dependent variant of C-index that truncates pairs within the prespecified time point Uno et al. (2011). The time dependent concordance index at $t$, $C^{td}(t)$, is defined as

$$\frac{\sum_{i=1}^{N} \sum_{j=1}^{N} e_i \{\hat{G}(T_i)\}^{-2} I(T_i < T_j, T_i < t) I(\hat{S}(t \mid \boldsymbol{x}_i) < \hat{S}(t \mid \boldsymbol{x}_j))}{\sum_{i=1}^{N} \sum_{j=1}^{N} e_i \{\hat{G}(T_i)\}^{-2} I(T_i < T_j, T_i < t)}$$

To evaluate $C^{td}$ at $[t_1, \ldots, t_L]$ at the same time, we take its mean $mC^{td} = \frac{1}{L} \sum_{l=1}^{L} C^{td}(t_l)$.

### A.1.2    TIME DEPENDENT AREA UNDER CURVE (AUC)

is an extension of the ROC-AUC to survival data Hung & Chiang (2010). It measures how well a model can distinguish individuals who fail before the given time ($T_i < t$) and who fail after the given time ($T_j > t$).

The AUC at time $t$, $AUC(t)$, is defined as

$$\frac{\sum_{i=1}^{N} \sum_{j=1}^{N} I(T_j > t) I(T_i \leq t) \omega_i I(\hat{S}(t \mid \boldsymbol{x}_i) \leq \hat{S}(t \mid \boldsymbol{x}_j))}{(\sum_{i=1}^{N} I(T_i > t))(\sum_{i=1}^{N} I(T_i \leq t) \omega_i)}$$

To evaluate $AUC$ at $[t_1, \ldots, t_L]$ at the same time, we take its mean $mAUC = \frac{1}{L} \sum_{l=1}^{L} AUC(t_l)$.

### A.1.3 Negative Binomial Log-Likelihood

We can evaluate the negative binomial log-likelihood (NBLL) to measure both discrimination and calibration performance Kvamme et al. (2019). The negative binomial log-likelihood at $t$ measures how close the survival probability is to 1 if the given data survived at $t$ and how close the survival probability is to 0 if the given data failed before $t$. The NBLL at $t$, $NBLL(t)$, is defined as

$$-\frac{1}{N}\sum_{i=1}^{N}\left[\frac{\log(1-\hat{S}(t\mid \boldsymbol{x}_i))I(T_i \leq t, e_i = 1)}{\hat{G}(T_i)} + \frac{\log \hat{S}(t\mid \boldsymbol{x}_i)I(T_i > t)}{\hat{G}(t)}\right]$$

For the convenience of evaluation, we integrate the NBLL, $iNBLL = \frac{1}{t_2-t_1}\int_{t_1}^{t_2} NBLL(t)dt$.

### A.2 Experimental Details

Across all datasets, we split training set, validation set and test set into 70%, 10% and 20% each using `PyTorch`'s `random_split` function Paszke et al. (2019). We set `seed = 42` when splitting.

Across all experiments, we use an MLP with two hidden layers where each layer has 64 hidden units. Across all layers, we apply Layer normalization Ba et al. (2016). Instead of naively feeding time $t$ into the neural network, we feed scaled time $\tilde{t} = (t - t_2)/(t_3 - t_1)$ where $t_1, t_2$, and $t_3$ are first, second, and third quartile of failure event times. We found that this strategy enhances the ODE model's performance and boosts training time. To incorporate time $t$ into the survival analysis model, we project the time into an eight dimensional vector using a single layer MLP and then concatenate it to the input data. The time $t$ is also specified by adding projected output into each layer output. We use the AdamW optimizer Loshchilov & Hutter (2019) and clipped the gradient norm so that it does not exceed 1. We set the learning rate to 0.001. We have implemented the code using `JAX` Bradbury et al. (2018) and `Diffrax` Kidger (2021) [9].

To find the best $\lambda$ in equation 7, we run experiments with $\lambda = 1, 5, 10, 50$ and report the results at $\lambda = 10$ as it shows decent performance across all metrics and datasets. We also have to set the number of samples $M$ from the time sampling process $t \sim p_\theta(t \mid \boldsymbol{x})$ in equation 5. We set $M = 5$ across all the hazard gradient penalty experiments. To find the best coefficient $\alpha$ in *ODE + L1*, *ODE + L2*, and *ODE + LCI* experiments, we set $\alpha = 1e-1, 1e-2, 1e-3$ and run the experiments. We report the best $\alpha$ in terms of $mAUC$. To report $mC^{td}$ and $mAUC$, we calculate $C^{td}$ and $AUC$ at 10%, 20%, ..., 90% event quantiles and average them. To report $iNBLL$, we integrate from the minimum time of the test set to the maximum time of the test set. We use `scikit-survival` Pölsterl (2020) to report $mC^{td}$ and $mAUC$. We use `pycox` Kvamme et al. (2019) to report $iNBLL$. Across all experiments, we run 7 experiments with different seeds and report their mean and the standard deviation.

## B Code Snippet of HGP

```
1  import jax
2  import jax.random as jrandom
3  import jax.nn as jnn
4  import jax.numpy as jnp
5
6  odeint = get_odeint(hazard_func, args.rtol, args.atol)
7
8  @jax.jit
9  def calc_dh_dx(
10     params: optax.Params, t: jnp.ndarray, X: jnp.ndarray
11 ):
12     def sum_h_func(params, t, X):
13         zeros = jnp.zeros((X.shape[0], 1))
14         states_t0 = jnp.concatenate((X, zeros), axis=-1)
15         return jnp.sum(hazard_func(t, states_t0, args=params)[:, -1])
```

---

[9]The code will be made publicly available in the near future.

```
16
17      return jax.grad(sum_h_func, argnums=2)(params, t, X)
18
19  @jax.jit
20  def loss_func(
21      params: optax.Params, X: jnp.ndarray, event: jnp.ndarray,
22      t: jnp.ndarray, timestamp: jnp.ndarray, unique_idx: jnp.ndarray,
23      key: jnp.ndarray
24  ):
25      batch_size = X.shape[0]
26      time_size = jnp.size(timestamp)
27
28      timestamp = jnp.insert(timestamp, 0, 0)
29      int_hazard_t0 = jnp.zeros((X.shape[0], 1))
30      states_t0 = jnp.concatenate((X, int_hazard_t0), axis=-1)
31      ys = odeint(timestamp, states_t0, params)
32      int_hazard = ys[1:, :, -1]
33
34      log_surv = - int_hazard
35      log_surv_t = log_surv[unique_idx, jnp.arange(X.shape[0])]
36      hazard_t = hazard_func(t, states_t0, args=params)[:, -1]
37
38      log_prob_t = jnp.log(hazard_t + 1e-6) + log_surv_t
39
40      assert log_surv_t.shape == log_prob_t.shape
41      assert event.shape == log_prob_t.shape
42
43      nll = - (event * log_prob_t + (1. - event) * log_surv_t).mean()
44
45      # now [batch_size, time_size]
46      logits = jnp.transpose(jax.lax.stop_gradient(log_surv), (1, 0))
47      time_idx = jrandom.categorical(
48          key, jnp.tile(logits[:, :, None], (1, 1, args.sample_size)),
49          axis=1
50      )
51
52      t0 = timestamp[0:][time_idx]
53      t1 = timestamp[1:][time_idx]
54      t_prime = t0 + (t1-t0) * jrandom.uniform(key, shape=t0.shape)
55      assert t_prime.shape == (batch_size, args.sample_size)
56
57      # E_{S(t | x)} || \nabla_x h(t | x) ||
58      dh_dx = jax.vmap(calc_dh_dx, in_axes=(None, 1, None))(params, t_prime, X)
59      assert dh_dx.shape == (args.sample_size, batch_size, X.shape[1])
60      dh_dx_norm = jnp.linalg.norm(dh_dx, ord=2, axis=-1).mean()
61
62      loss = nll + args.lambda_ * dh_dx_norm
63
64      return loss, (nll, dh_dx_norm)
```

## C  THE EXISTENCE OF THE PROBABILITY DISTRIBUTION $s_\theta(t \mid \boldsymbol{x})$

In general, we cannot guarantee the existence of probability distribution

$$s_\theta(t \mid \boldsymbol{x}) = \frac{S_\theta(t \mid \boldsymbol{x})}{\int S_\theta(t \mid \boldsymbol{x})dt}$$

as the integration $\int S_\theta(t \mid \boldsymbol{x})dt$ may not exist.

To ensure the existence of $\int S_\theta(t \mid \boldsymbol{x})dt$, we simply add a constraint: $h_\theta(t \mid \boldsymbol{x}) \geq \epsilon$. Here, $\epsilon$ is a very small constant (e.g. $\epsilon = 1e-8$). As $\epsilon$ is a very small constant, it has a negligible impact on the algorithm. The constraint can be achieved easily by adding $\epsilon$ to the softplus output of the hazard function.

Table 5: Comparison against STEER.

(a) $mC^{td}(\uparrow)$

| Method | SUPPORT | METABRIC | RotGBSG |
|---|---|---|---|
| ODE | $0.771 \pm 0.003$ | $0.695 \pm 0.008$ | $0.718 \pm 0.005$ |
| STEER | $0.772 \pm 0.002$ | $0.699 \pm 0.010$ | $0.718 \pm 0.006$ |
| HGP | $\mathbf{0.775 \pm 0.004}$ | $\mathbf{0.702 \pm 0.009}$ | $\mathbf{0.723 \pm 0.006}$ |

(b) $mAUC(\uparrow)$

| Method | SUPPORT | METABRIC | RotGBSG |
|---|---|---|---|
| ODE | $0.810 \pm 0.002$ | $0.729 \pm 0.005$ | $0.746 \pm 0.006$ |
| STEER | $0.810 \pm 0.001$ | $0.730 \pm 0.006$ | $0.745 \pm 0.006$ |
| HGP | $\mathbf{0.814 \pm 0.002}$ | $\mathbf{0.732 \pm 0.005}$ | $\mathbf{0.753 \pm 0.005}$ |

(c) $iNBLL(\downarrow)$

| Method | SUPPORT | METABRIC | RotGBSG |
|---|---|---|---|
| ODE | $0.516 \pm 0.015$ | $\mathbf{0.472 \pm 0.005}$ | $0.530 \pm 0.012$ |
| STEER | $0.521 \pm 0.012$ | $0.475 \pm 0.011$ | $\mathbf{0.523 \pm 0.006}$ |
| HGP | $\mathbf{0.506 \pm 0.011}$ | $0.479 \pm 0.003$ | $0.530 \pm 0.003$ |

Table 6: The taken time (in seconds) and the number of function evaluations (NFE) for each step in the training/evaluation time on SUPPORT. The numbers in parenthesis indicate relative performance against vanilla ODE. HGP boosts training time (8% faster) and gives rise to 9% speedup on evaluation time.

| | ODE | ODE + L1 | ODE + L2 | ODE + LCI | ODE + HGP |
|---|---|---|---|---|---|
| train time | 0.3392 (1) | 0.2688 (0.79) | 0.2758 (0.81) | 0.3434 (1.01) | 0.3135 (0.92) |
| eval time | 0.0019 (1) | 0.0015 (0.77) | 0.0015 (0.78) | 0.0018 (0.95) | 0.0018 (0.91) |
| train NFE | 10.980 (1) | 9.214 (0.83) | 9.151 (0.83) | 10.973 (0.99) | 10.556 (0.96) |
| eval NFE | 8.580 (1) | 7.073 (0.81) | 7.073 (0.82) | 8.446 (0.98) | 7.988 (0.93) |

Under the constraint, $S_\theta(t \mid \boldsymbol{x})$ is bounded by an exponential function:

$$S_\theta(t \mid \boldsymbol{x}) = \exp\left(-\int_0^t h(\tau \mid \boldsymbol{x})d\tau\right) \leq \exp(-at)$$

Also, $\int S_\theta(t \mid \boldsymbol{x})dt$ is bounded:

$$\int S_\theta(t \mid \boldsymbol{x})dt \leq \int \exp(-\epsilon t)dt = \frac{1}{\epsilon}$$

# D  COMPARISON AGAINST NEURAL ODES SPECIFIC REGULARIZER

In this section, we compare HGP against Neural ODEs specific regularizer: STEER (Ghosh et al., 2020). STEER regularizes Neural ODEs by perturbing the final time of the integration. Table 5 compares the performance of HGP against STEER. Overall, HGP outperforms STEER on various setups.

# E  THE TAKEN TIME AND THE NUMBER OF FUNCTION EVALUATIONS

In this section, we compare HGP against competing regularizers in terms of the taken time and the number of function evaluations on SUPPORT dataset. See Table 6 for the details. Overall, the result aligns with that of Table 3.

