# OpenReview forum: "Hazard Gradient Penalty for Survival Analysis"
_ICLR.cc/2023/Conference — Submitted to ICLR 2023_

### Official Review · Reviewer_pnuq · 2022-10-19

**Confidence:** 4
**Correctness:** 3
**Technical Novelty And Significance:** 2
**Empirical Novelty And Significance:** Not applicable
**Recommendation:** 3

**Clarity, Quality, Novelty And Reproducibility:**

The clarity is okay. The method is new. The method seems to be easy to implement.

**Strength And Weaknesses:**

Strength:
1. The authors propose a novel hazard gradient penalty specifically on survival analysis problems using ODE methods.

Weakness:
1. The use of the hazard gradient penalty is not well-supported.
a. The authors claim that
"The cluster assumption from semi-supervised learning states that the decision boundaries should not cross high-density regions (Chapelle et al., 2006). In a similar vein, hazard functions of survival analysis models should change slowly in high-density regions." I am not sure how the idea of cluster assumption applies here in survival analysis. It does not sound similar to me.
b. The authors somehow prove that their gradient penalty is an upper bound on KL-divergence between p(t|x) and p(t|x'). There are many ways to penalize that. I am not sure why the authors care about the gradient of the hazard. You can actually directly penalize the gradient of p(t|x). You can also model the CDF as an ODE[1] and do not need to model the cumulative hazard. You probably also want to cite [1] since they also talk about using ODE in survival analysis and is very similar to SODEN.
2. Empirical results are not very convincing. The authors do not tune much for the baseline ODE model. But it's actually sensitive to many hyperparams. See the SODEN repo. They random search 100 groups of hyperparams, select the best on the validation and report on the test. If the authors can show that their method is better than the numbers reported in SODEN, it will be more convincing empirically. Some large real world experiments may also be needed (for example, MIMIC, SEER). The current datasets are not very large and do not contain many features which may not be the right test-bed for deep learning.

[1]Derivative-based neural modelling of cumulative distribution functions for survival analysis.


**Summary Of The Paper:**

The authors propose a hazard gradient penalty on ODE models to ensure the smoothness on the density in survival analysis. The main contribution is this hazard gradient penalty.

**Summary Of The Review:**

It's unclear whether we should use the proposed method. The motivation from ssl is vague and there is obviously another way to penalize the function based on the theoretical claim. So I give "reject".

---

> ### Author Response · Authors · 2022-11-11
> **Response to the Reviewer pnuq**
>
> Thank you for your time and constructive feedbacks. We list the answers below to clarify your concerns.
>
> > I am not sure how the idea of cluster assumption applies here in survival analysis.
>
> We added a paragraph (in blue) to describe the necessity of cluster assumption in survival analysis in Section 3.
> We hope you find it helpful.
>
> We copy the paragraph here for your convenience.
>
> **Consider a case where two data points $x_1, x_2 \in \mathbb{R}^d$ failed at $t_1, t_2 (t_1 < t_2)$ each.
> Under the cluster assumption, a point $x_2^\prime \in B(x_2, \epsilon)$
>  should fail at $t_2^\prime \approx t_2$.
> If $\hat{p}(t \mid x_2^\prime)$ is skewed for some reason and puts high density at $t_1^\prime < t_1$, it worsens the model’s performance.
> To evade such situation,  $\hat{p}(t \mid x_2^\prime)$ should not deviate too much from $\hat{p}(t \mid x_2)$.**
>
> > You can actually directly penalize the gradient of p(t|x)
>
> Directly penalizing the gradient of $p(t \mid x)$ is **computationally inefficient (or intractable)**.
>
> $\nabla_x p(t \mid x) = \nabla_x h(t \mid x) S(t \mid x) = \nabla_x h(t \mid x) + \nabla_x S(t \mid x) = \nabla_x h(t \mid x) + \nabla_x \exp (\int - h (\tau \mid x) d \tau ) $
>
> Hence, to calculate $\nabla_x p(t \mid x)$, you need to calculate the gradient of integration.
> Calculating $\nabla_x p(t \mid x)$ during training will lead to very slow learning.
>
> On the other hand, our hazard gradient penalty is computationally much more efficient than directly minimizing $KL(p(t \mid x) \Vert p(t \mid x^\prime))$ or $\nabla_x p(t \mid x)$.
>
> > You can also model the CDF as an ODE[1] and do not need to model the cumulative hazard.
>
> The choice of modeling the cumulative hazard is **not our main concern**.
> The hazard gradient penalty can be applied to any survival analysis model as long as it models hazard function.
> Anyways, thanks for putting [1] into our attention, we cited the paper in the updated manuscript.
>
> > Empirical results are not very convincing.
>
> We added the experimental results of DeepHit.
> The performance gap (in terms of $iNBLL$) of DeepHit and ODE in our paper is similar to that in SODEN paper.
> We hope this ease your concern.
>
> > The current datasets may not be the right test-bed for deep learning.
>
> We do not agree. **SUPPORT and METABRIC have been test-beds for deep learning based survival analysis for years**.
> Statistically significant performance gains on SUPPORT and METABRIC **clearly shows the competence of the hazard gradient penalty**.
>
>
> We hope these answers clarifies your concerns.
>
> [1] Derivative-based neural modelling of cumulative distribution functions for survival analysis.

---

### Official Review · Reviewer_CgBz · 2022-10-23

**Confidence:** 4
**Clarity, Quality, Novelty And Reproducibility:** See above.
**Correctness:** 3
**Technical Novelty And Significance:** 2
**Empirical Novelty And Significance:** 2
**Recommendation:** 5

**Strength And Weaknesses:**

**Strengths**:
- The paper focuses on survival analysis, an important problem across many domains
- Theoretical analysis of connections of the hazard gradient penalty to KL divergence  minimization of  KL ($p(t|x) || p(t| x^{\prime})$), where $x^{\prime}$, may be of interest to some researchers
- Paper provides a snippet code implementation in JAX

**Weaknesses**:

*Overall Clarity*
-  The title of the paper needs to be more focused since the proposed regularizer  is mostly applicable to neural ODE survival models
- The paper should include a figure, illustrating (i) model parameters $\theta$; (ii) how $h_{\theta}(t|x)$ are learned, and (iii) model outputs at test time.

*Technical Quality*
- The paper claims modeling $p(t|x)$ or $S(t|x)$ is challenging:  This is not necessarily true, since there exist several approaches modeling $p(t|x)$ or $S(t|x)$ including AFT, Lee et al. (2018),  Chapfuwa et al. (2018), *etc.*
- It's unclear how the paper imposes the survival  constraints for learning valid distributions: Survival function is a monotonically decreasing function and $\lim_{t \rightarrow \infty } S(t|x) =0$
- It's unclear what the model outputs at test time.  E.g., Is the model predicting hazard, event times, or survival probabilities?
- Eqn 2: Should be $P(t < T < t + Δt | T> t, x)$
- Eqn 3: What are the inputs to the ODE solver?
- Paper claims modeling the hazard function is better that Tang et al. (2022b)’s cumulative hazard function without providing any justification, *e.g.*, experimental results
- Eqn 5: Should be $t \sim p_{\theta}(t|x)$, it's unclear how the paper samples from the survival function $t \sim S_{\theta}(t|x)$?
- Hazard gradient penalty: It's unclear why the ODESolver is evaluated at varying batch-specific times, instead of shared range $[0, t_{\rm max}]$
- Eqn 7: How is $\lambda$ selected?

*Underwhelming Experiments*
- The performance improvements over the vanilla neural ODE baseline seem marginal:
*  How significant are the claimed improvements
* Table 3: Computational complexity analysis is only provided for one dataset. How does the proposed approach scale with larger datasets (*e.g.*, Support), batch sizes, *etc.*?
-  Paper should also include qualitative results comparing predicted $h(t|x), S(t|x), p(t|x)$ against ground truth times and baselines
- For completeness, the paper should consider comparisons to other non-ODE baselines, e.g., AFT, Lee et al. (2018),  Chapfuwa et al. (2018), etc.
- Table 2: It seems the model performance is not sensitive to increasing from $M=1$ to $M=10$. Is this expected?

*Novelty*
- Gradient penalty regularisation has been proposed before. An expanded discussion highlighting how the proposed approach is better (or different) than previous approaches, *e.g.*, 1-Lipschitz function constraint in W-GAN is crucial.

**Summary Of The Paper:**

The paper proposes a hazard gradient penalty algorithm for regularising neural ode-based survival models. Additionally, the paper formulates a theoretical connection of the proposed hazard gradient penalty to KL divergence  minimization of  KL ($p(t|x) || p(t| x^{\prime})$), where $x^{\prime}$ are neighbouring data points of $x$. Experimental results on three real-world datasets show performance improvement over baselines per metrics C-Index, negative binomial log-likelihood (NBLL), and AUC.

**Summary Of The Review:**

Some researchers may find the theoretical analysis of the proposed gradient penalty interesting. However, the paper is lacking due to  several technical issues and underwhelming experiments.

---

> ### Author Response · Authors · 2022-11-11
> **Response to the Reviewer CgBz**
>
> Thank you for your time and constructive feedbacks. We list the answers below to clarify your concerns.
>
> ### Overall Clarity
>
> > The title of the paper needs to be more focused.
>
> As reviewer **ReJF** pointed out, we start from a very general form of survival analysis.
> As long as a model can model the hazard function, HGP can be applied.
> Hence we think that the title doesn't have to be changed.
>
> > The paper should include a figure, illustrating (i) model parameters $\theta$; (ii) how $h_\theta (t \mid x)$ are learned, and (iii) model outputs at test time.
>
> We didn't make figure illustrating model parameters, $h_\theta(t \mid x)$, and model outputs at test time as we follow conventional survival analysis models that model hazard functions.
> Nevertheless, you'll may find **the last paragraph of Section 2.1** and our **code snippet in JAX (Appendix B)** helpful.
>
> ### Technical Quality
>
> > The paper claims modeling $p(t \mid x)$ or $S(t \mid x)$ is challenging
>
> Yes, you are right. There are few works that directly models $p(t \mid x)$ or $S(t \mid x)$.
> However, to satisfy the constraints written in Section 2 (positivity, integrates to 1, monotonic decreasing function) the works have their own limitations.
>
> For example, Chapfuwa et al. (2018) let the model sample from $p(t \mid x)$ **at the cost of explicit probability distribution**. *i.e.* they can only sample from $p(t \mid x)$ but cannot access to the likelihood $p(t \mid x)$ as they introduced adversarial training scheme.
> Lee et al. (2018) also models $p(t \mid x)$ but they **discretized the time domain**.
>
> Tang et al. (2022b) also described the downside of survival analysis models that directly models $p(t \mid x)$ or $S(t \mid x)$ in their introduction.
>
> Nevertheless, we admit that the presentation is somewhat misleading and we changed the sentence into "Modeling $p(t \mid x)$ or $S(t \mid x)$ should satisfy the following constraints"
>
> > It's unclear how the paper imposes the survival constraints for learning valid distributions
>
> We model the hazard function using neural network followed by **softplus** activation function. This ensures $h(t \mid x) > 0$.
> Since we model $S(t \mid x) =\int_0^t - h(\tau \mid x)$, *i.e* integrate negative values ($-h(t \mid x)$), the monotonic decreasing property of $S(t \mid x)$ and $\lim_{t \to\infty} S(t \mid x) = 0$ is guaranteed.
>
> This is also described in **Section 2.1**. Please check.
>
> > It's unclear what the model outputs at test time.
>
> At test time, we predict survival probabilities through integrating the hazard function.
> However, in principle we can predict hazard, event times, and survival probabilities at test time.
>
> > Eqn 2: Should be $P(t < T < t + \Delta t \mid T > t, x)$
>
> As [1] and [2] describes $h(t \mid x) = \lim_{\Delta t \to 0} \frac{P(t \leq T < t + \Delta t \mid T \geq t, x)}{\Delta t}$, we will follow their description.
>
> > Eqn 3: What are the inputs to the ODE solver?
>
> Initial value 0 ($= \log S(0 \mid x)$), covariate x, the final time of integration $t_k$, and hazard function $h_\theta (t \mid x)$ are the inputs to the ODE solver.
>
> > Paper claims modeling the hazard function is better that Tang et al. (2022b)’s cumulative hazard function without providing any justification
>
> We are not claiming that our modeling is better than that of Tang et al. (2022b).
> We claim that depending the cumulative hazards is redundant if we follow the formulation of Equation 3.
> Tang et al. (2022b)'s formulation is also based on Equation 3.
> As our focus is the effectiveness of regularizers, we think that comparing our ODEs to that of Tang et al. (2022b) is **off the topic**.
>
> > Eqn 5: Should be $t \sim p_\theta (t \mid x)$, it's unclear how the paper samples from the survival function $t \sim S_\theta (t \mid x)$?
>
> This is described in **footnote 4** of the updated manuscript (footnote 3 of original version). Please check.
>
> > Hazard gradient penalty: It's unclear why the ODESolver is evaluated at varying batch-specific times, instead of shared range $[0, t_{max}]$
>
> Integrating from 0 to $t_{max}$ is a waste of computation as we don't need to access the hazard rate at $t_{max}$ and the survival probabilty at $t_{max}$.
> For efficient compuation, we integrate till $t_k$ where $t_k$ is the per-batch max time.
>
> > Eqn 7: How is $\lambda$ selected?
>
> We ran experiments with $\lambda = 1, 5, 10, 50$ and report the result at $\lambda = 10$ across all settings.
> The details are described in Appendix A.2
>
>
> [1] Kleinbaum, D. G., & Klein, M. (2012). Survival analysis: a self-learning text (Vol. 3). New York: Springer.
>
> [2] Katzman, J. L., Shaham, U., Cloninger, A., Bates, J., Jiang, T., & Kluger, Y. (2018). DeepSurv: personalized treatment recommender system using a Cox proportional hazards deep neural network. BMC medical research methodology, 18(1), 1-12.

---

> ### Author Response · Authors · 2022-11-11
> **Continuing the response**
>
> ### Underwhelming Experiments
>
> > For completeness, the paper should consider comparisons to other non-ODE baselines, e.g., AFT, Lee et al. (2018), Chapfuwa et al. (2018), etc.
>
> We added the experimental results of DeepHit.
>
> > The performance improvements over the vanilla neural ODE baseline seem marginal
>
> On most settings, the performance gain from HGP against vanilla ODE is **statistically significant**.
> Furthermore, given that the **performance gain by using ODE over DeepHit** is roughly 0.003 ~ 0.01 the **performance gain by using HGP over vanilla ODE** is by no means marginal.
>
> > Table 3: Computational complexity analysis is only provided for one dataset.
>
> Thanks for pointing this out. We added computational complexity analysis on SUPPORT in Appendix E (Table 6).
>
> > Paper should also include qualitative results
>
> We tried, but qualitative results are not so informative. If you have a good idea, please let us know.
>
> > Table 2: It seems the model performance is not sensitive to increasing from $M = 1$ to $M=10$. Is this expected?
>
> Usually, monte carlo estimates become more accurate when the number of sample size grows.
> However, there are some cases **where only one sample is enough**.
> For example, [1] argues that only one sample size was enough for estimating trace of a hessian using Hutchinson's trick.
>
> [1] Song, Y., Garg, S., Shi, J., & Ermon, S. (2020, August). Sliced score matching: A scalable approach to density and score estimation. In Uncertainty in Artificial Intelligence (pp. 574-584). PMLR.
>
> ### Novelty
>
> > Gradient penalty regularisation has been proposed before. An expanded discussion highlighting how the proposed approach is better (or different) than previous approaches
>
> You are right. Gradient penalty (GP) has been proposed in machine learning several times.
> We do not argue that we are the first to propose gradient penalty in machine learning.
> Our main contribution lies on the theoretical analyses of why GP is effective in survival analysis.
> The experimental results support our theoretical analyses.
>
> Nevertheless, we compare the performance of hazard gradient penalty against 1-Lipschitz constraint.
> The results are on SUPPORT dataset.
>
> |                   |       $mC^{td}$ |          $mAUC$ | $iNBLL$         |
> |------------------:|--------------:|--------------:|---------------|
> | ODE               | 0.771 (0.003) | 0.810 (0.002) | 0.516 (0.015) |
> | ODE + 1 Lipschitz | 0.698 (0.010) | 0.728 (0.009) | 1.349 (0.228) |
> | ODE+HGP           | 0.775 (0.004) | 0.814 (0.002) | 0.506 (0.011) |
>
> We report averages and standard deviations of 7 different runs.
> As the table shows, 1-Lipschitz constraint hinders learning.
> We conjecture that the 1-Lipchitz constraint may be **a too harsh constraint** for learning.

---

### Official Review · Reviewer_ReJF · 2022-10-25

**Confidence:** 4
**Correctness:** 3
**Technical Novelty And Significance:** 2
**Empirical Novelty And Significance:** 2
**Recommendation:** 5

**Clarity, Quality, Novelty And Reproducibility:**

This paper presents the related work, the motivation of proposing hazard gradient, and the main steps of how to implement this method clearly.

This paper's main novelty is proposing using a penalty based on hazard function, while the main compuational framework inherits from Neural ODE (Chen et al 2018).

The theoritial proof in section 3.2 looks valid. The algorithm presented in Alogorithm 1 and Experiemntal Details A.2 add sense while the reviewer didn't try to programmingly reproduce the result.

The clarity can be improved in a few detailed places, for example:
* Section 2: There are numerious splicit survival models. Tang 2022b adopted one form depending on cumulative hazard. The author of this under reviwe paper chose to work under more general survival regression form. Maybe the language "modeling p(t|x) or S(t|x) is challenging" and the footnote can be reworded.
* Section 4: 1) Specify "our proposed method" is called as ODE+HGP. 2) refer reader to A1 when first metion mC, mAUC and iNBLL. 3) Clarify In table 1 it's mean +/- what is shown, and mention some experimental details in A2.


**Details Of Ethics Concerns:**

Found this under review paper with author names listed under
https://arxiv.org/abs/2205.13717
https://deepai.org/publication/hazard-gradient-penalty-for-survival-analysis
https://www.researchgate.net/publication/360936267_Hazard_Gradient_Penalty_for_Survival_Analysis


**Strength And Weaknesses:**

This paper started from a very general form of survival analyses, making the proposed method applicable to broad survival models shown in section 2.1. The alogorithm proposed is computationally efficient than KL convergence by requiring less derivation of intermediant values. Theoretical proof of the equivalence of KL divergence and the hazard gradient method is proved. The experiments performed on three public datasets showed the proposed hazard gradient penalty method perform good under the three adopted criterions. The reviewer like the idea in A2 of feeing scale time to boost training time.

In A2, it clarifies "calculate C, and AUC at 10%, 20%, ..., 90% event quantiles and average them". This reviewer is wondering the comparision of the proposed method and existing one would be like what if closer quantiles (say each 1% apart), or if higher weight to cetral quantiles. Or say, would it possible the proposed method with estimated hazard change slow in boundary to make calculated mAUC etc look good while not better performance than existing penalties at the central high-density regions.


**Summary Of The Paper:**

This paper applied the neural ODE method to log trasformed survival rate model expressed by integration of hazard funciton. The penalty proposed is hazrd gradient penalty. The equivalence to KL divergence is proved, and empiral comparison to four other penalties under three criterions demonstrated the benifit of the proposed hazard gradient penalty.


**Summary Of The Review:**

This paper proposed hazard gradient penalty to address survial analyses using ODE framework facusing on data point rather than global information. The general presentation is good. The experimental result (if the author can clarify/defend the result with more than only ten quantiles selected) compared to existing method based on the paper showed advantage of implementing this regularizer.

---

> ### Author Response · Authors · 2022-11-11
> **Response to the Reviewer ReJF**
>
> Thank you for your time and constructive feedbacks. We list the answers below to clarify your concerns.
>
> > Evaluation of closer quantiles.
>
> We show $mC^{td}$ and $mAUC$. This time we calculate $C^{td}$ and $AUC$ at 10%, 11%, 12% ... 88%, 89%, 90% quantiles (high weight on central high density regions) and average them.
>
> **SUPPORT**
> |         |       $mC^{td}$ |          $mAUC$ |
> |--------:|--------------:|--------------:|
> | ODE     | 0.771 (0.002) | 0.812 (0.002) |
> | ODE+HGP | 0.775 (0.004) | 0.816 (0.002) |
>
> **METABRIC**
> |         |       $mC^{td}$ |          $mAUC$ |
> |--------:|--------------:|--------------:|
> | ODE     | 0.693 (0.007) | 0.727 (0.005) |
> | ODE+HGP | 0.700 (0.008) | 0.731 (0.006) |
>
> **RotGBSG**
> |         |       $mC^{td}$ |          $mAUC$ |
> |--------:|--------------:|--------------:|
> | ODE     | 0.718 (0.005) | 0.747 (0.006) |
> | ODE+HGP | 0.723 (0.005) | 0.754 (0.006) |
>
> We report averages and standard deviations of 7 different runs.
> Though the results are slightly different, the tendency does not change.
>
> > Maybe the language "modeling p(t|x) or S(t|x) is challenging" and the footnote can be reworded.
>
> Yes we do know that Tang2022b models $p(t \mid x)$ and $S(t \mid x)$. However, they do not model $p(t \mid x)$ and  $S(t \mid x)$ **directly**.
> What we wanted to say was: 'Due to the constraints that should be satisfied, modeling $p(t \mid x)$ and $S(t \mid x)$ directly is challenging and it is easier to model hazard function and then infer $p$ and $S$ through the hazard function.'
>
> Of course there are several researches that directly models $p(t \mid x)$ or $S(t \mid x)$.
> However, as written in Tang2022b, they have their own limitations.
> For example, AFT [1] makes parametric assumptions and DeepHit [2] discretizes time domain.
>
> Nevertheless, we admit that the presentation is somewhat misleading.
> The sentence was re-phrased into "Modeling $p(t \mid x)$ or $S(t \mid x)$ should satisfy the following constraints" in the updated manuscript.
>
> > Miscellaneous fixes in Section 4
>
> Thanks for correcting. We fixed these and colored them blue.
>
> [1] Wei, L. J. (1992). The accelerated failure time model: a useful alternative to the Cox regression model in survival analysis. Statistics in medicine, 11(14‐15), 1871-1879.
>
> [2] Lee, Changhee, William Zame, Jinsung Yoon, and Mihaela Van Der Schaar. "Deephit: A deep learning approach to survival analysis with competing risks." In Proceedings of the AAAI conference on artificial intelligence, vol. 32, no. 1. 2018.

---

### Official Review · Reviewer_T7HL · 2022-10-30

**Confidence:** 2
**Correctness:** 3
**Technical Novelty And Significance:** 3
**Empirical Novelty And Significance:** 3
**Recommendation:** 6

**Clarity, Quality, Novelty And Reproducibility:**

The paper is well-written and the authors have made their contributions clear. This work has novelty and was an interesting read. The work seems fairly reproducible given the provided code snippet by the authors.

**Strength And Weaknesses:**

* A major short-coming of this work is being restricted to time-static covariates. There has been recent work on time-varying covariates in survival analysis so considering time-static covariates only comes at the price of usability.


* Since you are proposing a hazard estimator, it would have been beneficial to also provide RMSE results on recovering the true hazard in a dataset. Of course for that you needed to create a synthetic dataset where you simulate survival data using a known hazard function but it would have been beneficial.


* The only non-ODE-solver baseline was CoxPH, whose dependence on input covariates is linear. Comparing against non-linear hazard estimators or even CoxPH extensions whose covariate effect is non-linear would have made the a stronger case of the developed algorithm. Without the mentioned comparisons, only ODE solvers are compared with one another.

**Summary Of The Paper:**

ODEs have recently found their applications in survival analysis where in the case of right-censoring the log-likelihood can be optimized quite straightforwardly using an ODE solver. This work is inspired by an assumption known the clustering literature according to which the decision boundaries do not cross high-density regions in the data space. They use the same intuition in their hazard estimation. They propose a regularizer on the hazard gradient which is the expectation of the hazard gradient with respect to a random variable whose PDF is the normalized survival function. They show that it is equivalent to minimizing the KL divergence between the PDFs of two input vectors which are "close" in the input space.

**Summary Of The Review:**

I find this paper interesting but at the same time I am not fully convinced of its applicability and superiority over baselines. Not sure about applicability because it only incorporates time-static covariates and not sure about superiority given the missing comparisons mentioned above.

---

> ### Author Response · Authors · 2022-11-11
> **Response to the Reviewer T7HL**
>
> Thank you for your time and positive feedbacks. We list the answers below to clarify your concerns.
>
> > This work is being restricted to time-static covariates.
>
> You are right. It is not straightforward to apply HGP to time-varying covariates.
> Applying HGP to time-varying covariates is a promising future direction.
>
> > Comparing against non-linear hazard estimators
>
> We added the experimental results of DeepHit [1] in Table 1. Though DeepHit does not model hazard function directly, as it utilize non-linear model to estimate first hitting time, it can be a competitive baseline.
>
> [1] Lee, Changhee, William Zame, Jinsung Yoon, and Mihaela Van Der Schaar. "Deephit: A deep learning approach to survival analysis with competing risks." In Proceedings of the AAAI conference on artificial intelligence, vol. 32, no. 1. 2018.

---

### Author Response · Authors · 2022-11-17
**Response to all reviewers**

We thank the reviewers for their time and constructive feedback. We would like to notice that the reviewers acknowledged the different **strengths** of our work.

## Strengths

- Novelty (Reviewer **T7HL, ReJF, pnuq**)
- Clarity (Reviewer **T7HL, ReJF, pnuq**)
- Well written (Reviewer **T7HL**)
- Reproducibility (Reviewer **T7HL, ReJF, CgBz**)
- Theoretical Analysis (Reviewer **ReJF, CgBz**)

The reviewers also raised several concerns. We believe our answers below will address the concerns.
We also added several corrections (blue-colored texts).

We would be happy to engage in further discussions raised by the reviewers.

---

### Decision · Program_Chairs · 2023-01-20

**Decision:**

Reject

**Justification For Why Not Higher Score:**

Reported marginal improvements over of baselines over limited set of benchmarks, limited novelty of some aspects of the contribution.
Most reviewers vote to reject.

**Justification For Why Not Lower Score:**

n/a

**Metareview: Summary, Strengths And Weaknesses:**

This easy to follow paper has been assessed by four knowledgeable reviewers. It presents a novel approach with a somewhat unclear or limited utility and unclear comparison, in terms of empirical performance, to state of the art alternatives. One of the reviewers correctly points out lack of novelty of the proposed gradient penalty regularization scheme. The reviewers also point out that the reported performance improvements over basic alternatives appear marginal. The choice of benchmark datasets is limited and blurs the perception of impact.
Three of the reviewers ranked this work at its current stage below the threshold of acceptance, one considered marginally acceptable. Therefore, I recommend rejection, but I would also like to encourage the authors to address the key deficiencies and resubmit their updated work to a reputable venue.